# Fatigue-Crack Detection in a Multi-Riveted Strap-Joint Aluminium Aircraft Panel Using Amplitude Characteristics of Diffuse Lamb Wave Field

**DOI:** 10.3390/ma16041619

**Published:** 2023-02-15

**Authors:** Frank H. G. Stolze, Keith Worden, Graeme Manson, Wieslaw J. Staszewski

**Affiliations:** 1Department of Mechanical Engineering, Sheffield University, Mappin St., Sheffield S1 3JD, UK; 2Department of Robotics and Mechatronics, AGH University of Science and Technology, Al. Mickiewicza 30, 30-059 Krakow, Poland

**Keywords:** fatigue crack, aircraft structures, crack detection, Lamb waves

## Abstract

Structural health monitoring of riveted aircraft panels is a real challenge for maintenance engineers. Here, a diffused Lamb wave field is used for fatigue-crack detection in a multi-riveted strap-joint aircraft panel. The panel is instrumented with a network of low-profile surface-bonded piezoceramic transducers. Various amplitude characteristics of Lamb waves are used to extract information on fatigue damage. A statistical outlier analysis based on these characteristics is also performed to detect damage. The experimental work is supported by simplified modelling of wave scattering from crack tips to explain complex response features. The Local Interaction Simulation Approach (LISA) is used for this modelling task. The results demonstrate the potential and limitations of the method for reliable fatigue-crack detection in complex aircraft components.

## 1. Introduction

It is well known that acquisition and running costs are critical in air transportation. Aircraft design relies heavily on the safe-life design concept, which assumes that all major components have no margin for failure. The safe operating life of these components is guaranteed by selecting large margins of safety. As a consequence, this overdesign approach generally inflicts high acquisition and running costs and may lead to poor performance. On the contrary, the most cost-effective design concept—the damage-tolerant concept—assumes that structural damage is inevitable. As a consequence, reliable damage detection and monitoring techniques are required. The solution can be offered by Structural Health Monitoring (SHM), which aims to provide nearly real-time, reliable information on structural integrity, as explained in [1,2,3,4,5]. This approach requires a network of sensors that are integrated with monitored structures. SMART Layers^®^ [6] are good examples of transducers that can be used in various network configurations for damage detection applications based on ultrasonic waves. SMART Layers^®^ comprise small piezoceramic transducers on a thin dielectric Kapton layer which can be easily surface-mounted on a structure or integrated into a structure/material.

Various methods based on ultrasonic waves have been developed for damage detection in structures. Guided ultrasonic waves, such as Lamb waves, have been considered for damage detection applications in aircraft components for many years [1,3,7]. A good summary of physical principles exploited for Lamb wave generation and sensing, respective transducer designs, different damage detection strategies and application examples, are given in [8,9,10,11]. All major Lamb-wave-based techniques used for damage detection can be classified into pulse-echo and pitch-catch methods [7,11]. The former methods use a narrowband excitation that is applied to monitored structures. Damage detection and monitoring then rely on signals reflected from damage. The latter approaches also utilise narrowband excitation, but in contrast to pulse-echo methods, Lamb wave signals interrogating monitored structures are transmitted through damage. These transmitted Lamb wave responses are then used for damage detection and monitoring. Often wave attenuation, phase shifts and mode conversion are used to obtain information about damage detection. It appears that the majority of Lamb wave applications for damage detection are related to fatigue-crack detection in metallic plate-like structures. Dispersion characteristics are easy to obtain in such structures. Wave propagation phenomena (e.g., reflections, scattering, attenuation) that are used for damage detection are also not difficult to model and analyse. Therefore, the majority of applications rely on carefully selected Lamb wave modes and the well-understood physics of wave propagation. Lamb wave propagation in composite structures always introduces additional complexity, but does not introduce any major difficulty, and numerous application examples can be found in the literature. Damage detection in adhesively bonded or sealed lap joints—that are often geometrically complex—is much more difficult. Nevertheless, damage detection in such structures has been addressed in research investigations. Important theoretical, numerical and experimental investigations that give interesting insight into the problem of lap-joint monitoring include the following research studies: bond-inspection methods for monitoring of tear straps and lap splice joints [12,13], mode-selection analysis for damage detection in adhesively bonded lap joints [11,14], hidden crack detection in multi-layered structures [15] and analysis of boundary conditions in multi-layered metallic structures [16,17]. The methods used for damage detection in lap joints include well established Non-Destructive Testing (NDT) approaches, such as: ultrasonic testing [18], eddy current [19,20], acoustic emission [21,22] and thermography [23,24]. They also include more novel techniques, such as: electronic scanning-laser Doppler vibrometry [25], speckle-pattern interferometry [26], electromechanical impedance [27], the Comparative Vacuum Monitoring (CVM) technique [28], vibro-acoustic monitoring [29] or a combination of some of these techniques [30].

Riveted lap joints are of particular interest to aerospace engineers; their design and properties have been extensively studied and applied for decades [31,32]. Lamb-wave-based damage detection in riveted lap joints—often used in aircraft structures—are still limited, and include research studies related to the monitoring of: fatigue cracks [33,34], multi-crack growth [35,36,37], bond quality [38], missing rivets [30] and simulated cracks [7]. Damage visualisation has been investigated in [39,40,41]. Lamb wave scattering using analytical and finite element modelling approaches has also been investigated [42]. The probability of crack detection and the probabilistic fatigue life of aircraft riveted lap joints has been studied in [43]. These applications mainly relate to simple laboratory specimens. Application to full-scale, real engineering structures are very limited in this field and include [44,45,46].

Monitoring such structures is much more challenging. Wave propagation modelling is very difficult because of material interfaces, differences in material impedances, varying thickness, reflections from geometrical boundaries (from rivets in particular), increased levels of attenuation or the modelling complexity of adhesives [14]. As a result, identification of Lamb wave modes and different types of reflected wave components (from geometrical features and damage) is not possible, since all these components are superimposed and overlapped. Therefore, the so-called diffuse Lamb wave field [47] is often analysed. This approach is concerned with a global analysis of wave responses that include all possible reflections from various geometrical features and damage. Since, in riveted lap joints, reflections from rivets are difficult to distinguish from reflections from cracks, pitch-catch damage detection methods are preferred when diffuse Lamb wave responses are analysed. In that case, subtracting baseline measurements representing undamaged conditions is a possible solution to the problem. Alternatively, soft or natural computing methods—reviewed in [48]—can be employed, as demonstrated in [49,50,51,52]. The former approach—i.e., subtracting baseline measurements—can be used on the assumption that operational and environmental conditions do not disturb the analysed wave field. Unfortunately, often signal changes produced by defects tend to be small compared with those associated with operational/environmental conditions and so are difficult to detect reliably. The latter approach—i.e., soft computing—is not always possible because of the significant data required for learning. Therefore, statistical analysis of Lamb wave responses—that involves less data—can be also employed [53].

In summary, two major approaches are used when Lamb waves are applied for damage detection. The first one relies on good physical understanding of the wave propagation field and often involves analytical, semi-analytical and/or numerical modelling in order to obtain dispersion curves. This approach is possible only for simple (laboratory) structures. The second approach assumes that full physical understanding of wave propagation fields for complex structures is not possible. Then the diffuse Lamb wave field is analysed using various machine learning approaches. The penalty in this approach is the large amount of data required. This is not always possible, particularly in real aerospace applications. In addition, despite numerous research efforts, the technological readiness of Lamb-wave-based damage detection methods for monitoring complex engineering structures—such as riveted lap joints—is still far from potential applications. It is clear that further research investigations are required to facilitate this development.

The objective and major novelty of this paper is to demonstrate that the third approach is possible. Even if a diffuse Lamb wave field is used, no large amount of data is needed and relatively simple data processing can be applied to extract useful crack detection/monitoring information.

This paper reports on Lamb-wave-based crack detection results from an aircraft multi-riveted lap joint panel. The work presented results from two series of very extensive experimental fatigue tests and crack detection tests conducted eight years apart. Interestingly, these tests involved different experimental equipment, operational and environmental conditions. Crack detection was based on the amplitude analysis of diffuse Lamb wave responses. The application of simple signal processing was intentional, to demonstrate that the monitoring strategy and access to good quality data from inspections performed over a long period of time is more important than sophisticated data analysis for successful damage detection. The focus was also on the performance of the sensors and their ability to detect a crack and to monitor its growth over a series of rivets. It is important to note that limited and relatively basic modelling work has been undertaken only to explain the investigated damage index. Such a comprehensive approach (i.e., complex specimen, extensive programme of experimental work, multi-sensor measurements, diffuse Lamb wave field analysis, sensor-quality assessment, comparative study of different amplitude-based damage indices, novelty detection and modelling undertaken to explain the results) to this difficult problem is also the novelty of the work presented.

This paper consists of four major parts. The multi-riveted strap joint aluminium panel and all the experimental arrangements associated with fatigue testing, measurements and crack detection are described in Section 2. Section 3 briefly presents three amplitude-based damage indices used for crack detection and monitoring. Lamb wave data quality assessment and crack detection results are presented in Section 4. Simple numerical simulations related to wave diffraction from a crack tip—presented in Section 5—complements the work presented, helping to explain some crack detection results. Discussion of the results and interesting observations related to practical monitoring aspects are given in Section 6. Finally, this paper is concluded in Section 7, which summarises the major work undertaken, gives major findings and proposes some ideas for future research investigations.

## 2. Experimental Arrangements

### 2.1. Multi-Riveted Strap Joint Aluminium Panel

The complex metallic specimen used for the fatigue experiment consisted of two symmetrical aluminium alloy multi-riveted plates that were connected through a strap joint with overall dimensions of 750 × 300 × 2 mm (Figure 1). The splice joint was fastened to either side by three rows of rivets, adding up to a total of 6 rows with 84 rivets. The specimen was specially designed and manufactured in such a way that it was representative of a standard aircraft splice joint. The only difference to a standard aircraft joint was the intentionally introduced 1.8 mm-long notches on either side of the central rivets by means of spark erosion. Furthermore, for the central rivets, slightly larger and stiffer rivets were used to ensure that cracks were initiated at those points.

### 2.2. Sensor and Measurement Arrangements

The specimen was instrumented with four SMART Layers^®^ from *Acellent Technologies, Inc.* (Sunnyvale, CA, USA). The SMART Layer^®^ is a thin dielectric film with embedded network of piezoelectric transducers that can act as actuators and sensors. The thin dielectric film was manufactured from graphite-epoxy with the [0//90/45/−45]_2*S*_ lay-up. The actual transducers have a low profile (thickness of 0.254 mm) in the shape of thin circular discs with a diameter of 6.35 mm. The transducer discs can operate in thickness and radial vibration modes. The frequency constants for the piezoceramic material—according to the manufacturer—are given as 2032 and 1980 Hz·m for the thickness and radial resonance modes, respectively. More physical parameters related to the SMART Layers^®^ can be found on the manufacturer’s webpage.

The SMART Layers^®^ were symmetrically aligned to the predicted crack propagation path, as shown in Figure 1. Altogether 12 transducer (actuator-sensor) pairs were used to monitor the upper and lower cracks. Figure 1b and Figure 2 label the transducer layers as A, B, C and D. The transducer layers A and B monitored the crack initiated by the upper central rivet while the layers C and D were arranged to monitor the crack emerging from the lower central rivet. The layers A and D were used for excitation, whilst the layers B and C were utilised for reception.

The wave propagation paths were described using layer labels (for the actuator and sensor), and the transducers’ numbers in each layer. Thus, the description A1-B6 indicates that a Lamb wave was generated by the actuator 1 from the transducer layer A and the response was sensed by the sensor 6 from the transducer layer B. Figure 2 illustrates 18 wave propagation paths investigated for the A and B layers. These paths were classified as CENTRAL (Figure 2a), LEFT (Figure 2b) and RIGHT (Figure 2c) to simplify the analysis of Lamb wave data. The CENTRAL wave propagation paths cross the central rivet where the crack was initiated. Another discrimination of all 18 acoustic paths can be made according to the 3 different wave propagation lengths involved. There were six direct paths with opposing transducers (e.g., A3–B3), with a distance of 75 mm, ten short diagonal paths (e.g., A3–B4), with a propagation length of 85 mm, and two long diagonal paths (A1–B6 and A6–B1), with a length of 213.6 mm.

### 2.3. Fatigue Tests and Crack Propagation

The panel was exposed to cyclic loading. The fatigue experiment consisted of two phases. The first experimental phase was performed in 2000 (PHASE I of the experimental test), as part of the EU MONITOR (Monitoring ON-line Integrated Technologies for Operational Reliability) project [49]. Lamb wave responses for 13 different damage severities (or cycle counts)—including the undamaged condition—were recorded for each respective transducer-pair combination. The final data acquisition was made at 137,500 cycles, when small cracks around the central rivet partially reached the first neighbour rivet. The same specimen was fatigue tested again after eight years (PHASE II of the experimental test). Lamb wave responses were acquired for 51 different damage severities. The experiment was terminated with a complete fracture of the specimen in the upper rivet line after 465,500 cycles, thereby utilising its entire fatigue lifetime.

The entire fatigue experiment was performed in a *Schenck* servo-hydraulic test machine. The panel was loaded statically with a tensile load of 22 kN and a sinusoidal dynamic load of ±18 kN at a frequency of 6 Hz, corresponding to a total load between 4 kN and 40 kN. The crack lengths were determined by using a fatigue-crack microscope featuring scales and a reticle. The cycling frequency was equal to 6 Hz when the PHASE I test was performed. This frequency was reduced to 1 Hz in the PHASE II test. The phases involved different clamping arrangements. A microscope was employed to precisely observe crack lengths.

Figure 3 shows the crack propagation curves. Here, the descriptions “upper crack” (blue line with squares) and “lower crack” (red line with circles) stand synonymously for the cracks in the upper (monitored by the A and B transducer layers), and lower (monitored by the C and D transducer layers) parts of the aluminium panel, respectively. The vertical line (described by the symbol “II”) separates the two experimental tests (PHASE I and PHASE II) investigated. The PHASE I test was completed after 137,500 cycles. Rivets located in the neighbourhood of the central rivet are labelled on the right-hand side vertical axis according to their order of appearance in the crack propagation paths. Only rivets that the crack intersected are described, i.e., the upper left crack propagated through the neighbour rivets 1, 2 and 3 before it left the overlapping area of the splice joint.

The crack measurements only present visible crack lengths on the reachable outer surface of the specimens. The vertical leaps in the crack length of Figure 3 only indicate that the respective cracks intersected rivets, leading to a sudden increase in the crack length, since the far side of the rivet hole with respect to the central rivet becomes suddenly relevant.

The crack lengths presented in Figure 3 show unambiguously that the PHASE I test was stopped when only the upper left crack reached the first neighbour rivet hole. Figure 3 demonstrates that the upper right crack required 22,000 load cycles more to reach its first neighbour rivet; it also shows that the crack lengths remained constant for a considerable number of cycles in the PHASE II test. This behaviour arose as the propagating cracks encountered rivet holes which have—because of their perfectly circular-shaped rivet diameter—a significantly lower crack intensity factor than the original crack. Thus, once a crack has reached a rivet, a new crack had to be initiated, hence slowing down and impeding the crack propagation. The crack pattern in a similar aircraft splice joint panel shown in [33] appears to be much more complex, with initiated cracks at numerous rivets at the same time compared to the almost linear pattern observed in this investigation. Only towards the end of the fatigue experiment were two parallel growing cracks noticed on the upper left side. This development was neglected in this study, as this additionally appearing crack near the rivet was of inferior importance to the actually relevant crack propagation outside the splice joint area. Hence, neither did it alter the fatigue lifetime of the specimen nor did it affect the recorded waveforms.

The vertical auxiliary lines in Figure 3 indicate where the left and right crack intersected the corresponding wave propagation paths. These vertical auxiliary lines will be shown in subsequent damage detection graphs, providing a visual aid for the performance analysis of the features relevant to damage detection.

The first nearly 5 mm crack was measured after 80,000 cycles. Figure 4a shows the photograph of the upper crack after 347,500 cycles, whereas Figure 4b gives the photographs of the fractured metallic panel after 465,500 cycles.

### 2.4. Lamb Wave Generation and Sensing

A burst signal consisting of five cycles of sine waves was used for Lamb wave generation. The amplitude of excitation was equal to 10 V. An excitation frequency of 410 kHz was applied for Lamb wave generation. Since a diffuse Lamb wave field was used for damage detection, this frequency was selected experimentally using a trial-and-error approach to generate responses with the best SNR. The excitation signal was generated using TTi *TGA1230* and *TGA1242* waveform generators in the PHASE I and PHASE II tests, respectively. The data acquisition was accomplished using a *LeCroy 9304 AM* and *LeCroy Waverunner LT264* oscilloscopes in the PHASE 1 and PHASE 2 tests, respectively. Signals were sampled at 25 MS/s in both tests. When the PHASE 1 test was performed each signal consisted of 2500 samples, giving the time span of 100 μs. The number of samples and time span were increased to 5000 and 200 μs, respectively, in the PHASE 2 test. Final responses for each investigated sensor path were averaged 10 and 50 times in the PHASE I and PHASE II tests, respectively. Data averaging was used to improve the Signal-to-Noise-Ratio (SNR). Signal responses were saved to the built-in disk drive of the oscilloscope and transferred to a PC for further analysis. Initial data acquisition tests were performed to show redundancy and repeatability, and hence to eliminate possible erroneously measured data sets. All Lamb wave signals were taken while the specimen was exposed to a static load of 22 kN in the servo-hydraulic test machine. Figure 5 gives an example of the excitation and response signals.

It is important to note that the eight-year period between the two tests was too long to keep exactly the same experimental conditions. For example, SMART Layer^®^ cable connections deteriorated and required some additional soldering and repair. Different clamping arrangements were used, as mentioned above. Damage detection experimental tests were performed inevitably in different temperature conditions that were not carefully controlled. It is well known that environmental conditions, particularly temperature, affect Lamb wave propagation, as demonstrated for example in [1,2,3,4,5]. However, all these problems were not such a bad thing altogether, and somewhat mimicked the possible operational conditions.

It is also worth mentioning that the piezoelectric elements in the SMART Layers^®^ were still operating normally when the PHASE II test was performed after almost 8 years of storage. This is rather impressive, considering the detrimental ageing effects, such as a potential deterioration of the coupling between the structure and the SMART Layers^®^ (bonding) and possible depolarisation of the piezoelectric material.

## 3. Signal Features Used for Damage Detection

The most common features used for damage detection are based on scattering, attenuation and mode conversion [1,2,3,4,5,8,9,10], changes in wave propagation velocity and variations in time-of-flight (TOF) of Lamb waves. Attenuation can be caused by several mechanisms, i.e., signal spreading because of dispersion and beam divergence, material damping, leakage into surrounding media and scattering [9]. However, only the latter allows one to make an inference of the occurrence of damage [8]. Damage detection methods based on attenuation in ultrasonic signals comprise a significant range of methods, as the attenuation can be reflected in the time, frequency and timescale domains. This paper utilises three different damage indices based on attenuation.

One of the simplest measures that can be used to assess attenuation is the trivial—but most commonly used—peak-to-peak value, defined as the difference between the maximum and minimum amplitudes of the signals, i.e.,
(1)$$DI={\left|max\left[y\left(t\right)\right]-min\left[y\left(t\right)\right]\right|}_{t_{2}}^{t_{1}}$$

The second damage index used was the maximum of the instantaneous amplitude, often called the envelope function. The instantaneous amplitude can be obtained by evaluating the magnitude of the *analytical signal* z (*t*), i.e.,
(2)$$\left|z\left(t\right)\right|=\sqrt{x{\left(t\right)}^{2}+\hat{{x\left(t\right)}^{2}}}$$
where $\hat{x\left(t\right)}$ is the Hilbert transform of *x*(*t*). Since the Hilbert-transform-based concept of instantaneous characteristics is valid only for narrow-band signals [1], smoothing based on low-pass filtering was applied to the envelope in order to remove possible spurious oscillations.

Finally, the integrated instantaneous amplitude was used as the third damage index. The idea behind integration was to remove any time dependency. The resulting quantity was comparable to the energy contained in the response signal.

## 4. Damage Detection Results

This section presents crack detection results. Firstly, the quality of the Lamb wave responses is assessed in terms of the SNR; various damage indices—based on the amplitude analysis—are then presented.

### 4.1. Noise Level Estimation

The direct relationship between propagation distance and attenuation means that the signal strength will be reduced with increasing wave propagation distance. Moreover, as noise is omnipresent in real measurements, the ratio between the signal and the noise will also reduce. In reality, measurements are always corrupted to some degree by noise. Therefore, the measurements *x*(*t*) can be conceived as noisy signals [54], and can be expressed as a superposition of the actual quantity of interest *s*(*t*) and noise *n*(*t*), i.e.,
(3)$$x\left(t\right)=s\left(t\right)+n\left(t\right)$$

In many contexts of signal processing, the information of only the signal strength or only the noise power is insufficient to describe the performance of certain methods. For this reason, it seems to be more appropriate to define the SNR level. Even though a measurement can be mathematically described using Equation (3), it is difficult in practice to separate the noise from the actual signal, as those components are not known *a priori*. Therefore, the SNR cannot easily be determined in a straightforward fashion. The SNR used in the current study is based on a maximum-likelihood estimate defined in [55]. This particular estimate seemed to be ideal for the problem analysed as it estimates the SNR from two versions of a Gaussian signal, corrupted by independent Gaussian noises. The estimation of the SNR from two versions of one signal, *x* and *y*, is thus given by,
(4)$$\mathrm{SNR}=\frac{2\;\sum_{i=1}^{N}x_{i}y_{i}}{\sum_{i=1}^{N}{\left(x_{i}-y_{i}\right)}^{2}}$$

Figure 6 gives a matrix of the SNR estimates for the number of measurements versus the utilised wave propagation path (transducer pair). Two different areas can be identified in Figure 6. Firstly, a generally darker area (mainly around 30 dB) comprising measurements from the beginning until the measurement number 13 for all transducer pairs, and secondly, all remaining measurements which appear generally in brighter colours (principally around 35 dB), hence exhibiting higher SNR. Those two areas demonstrate nicely the influence of the number of averages taken for one measurement, because only 10 averages built the basis for each measurement in the PHASE I test, while a recorded acoustic signal in the PHASE II test was the result of 50 averages. In other words, ensemble averaging is a simple and practical method to reduce the effect of noise in measured signals, and thus improves the SNR, as expected. Moreover, the results show that the transducer combinations A1B6 and A6B1 have low SNR—as indicated by a darker colour (horizontal bars)—compared to the SNR estimates of the remaining measurements.

### 4.2. Peak-to-Peak Amplitude

Figure 7a, Figure 8a and Figure 9a shows the results for the peak-to-peak amplitude calculated for all Lamb wave responses for the CENTRAL, LEFT and RIGHT wave propagation paths respectively, as defined in Section 2.2. The left column in these figures presents the results for the PHASE 1 experimental test (i.e., up to 137,000 fatigue cycles), whereas the right column gives the overall results for the PHASE I and PHASE II experimental tests (i.e., up to 465,500 cycles). The beginning of the PHASE II experimental test (after 137,500 cycles) is indicated by the black vertical line labelled with the Roman numeral “II”. The results are plotted versus load cycles. In order to uncover any correlation between the analysed damage index and crack length, two separate vertical axes are used. The blue ordinate axis on the left is dedicated to the peak-to-peak amplitude, and the green ordinate axis on the right describes the crack length. The black line with “+” symbols gives the crack propagation curve. Two sets of results for each analysed transducer pair are presented to give an indication of the variation of measurements when subsequent actuators are swapped with sensors. Additional numbers, i.e., 1 and 2, are attached to the transducer labels in the legends to account for these two data sets.

The results for the A3B4 and A4B3 wave propagation paths in Figure 7a monotonically decrease with increasing crack length. As the wave propagation length for the A1B6 and A6B1 paths is considerably longer (more than 2.5 times of A3B4 or A4B3), the attenuation of the waves increases, resulting in relatively low amplitude values (<±5 mV) in the waveforms, and hence low peak-to-peak values (<10 mV). For that reason, their respective damage indices can only show small reductions throughout the entire experiment. Those reductions can only be really noticeable towards the end of the experiment (i.e., after 340,000 cycles). After the experiment was resumed (after 137,500 cycles), a localised peak—that is difficult to explain—can be observed around 162,500 cycles for the A4B3 path. Nevertheless, the results for the A3B4 and A4B3 paths demonstrate a good correlation with damage severity, as they remain constant, following the constant size of the crack and reducing further as soon as the crack continued to grow (after approximately 340,000 cycles). However, this reduction is not monotonic, as small oscillations can be observed in this range of cycles. This particular behaviour is further examined in Section 5.

Unlike the graphs in Figure 7a, the majority of graphs in Figure 8a for the LEFT paths exhibit the expected constant values throughout the PHASE I experimental test. Only the characteristics for the A3B3 path diverge from this behaviour. These characteristics demonstrates considerably elevated peak-to-peak values between 80,000 and 95,000 cycles despite the continuously growing crack. However, after 100,000 cycles, the same characteristics exhibit an almost perfect linearly decreasing damage index, until the end of the PHASE I experimental test. This behaviour coincides with theoretical expectations, as the A3B3 path is supposed to indicate the occurrence of damage amongst the LEFT paths first. Furthermore, Figure 8a confirms the dependence between amplitude and the propagation length, as the direct paths AnBn (for n = 1,2 or 3), with the length of 75 mm, generally demonstrate higher peak-to-peak values compared to the diagonal paths with the length of 85 mm. In the transition from the PHASE I to the PHASE II experimental tests in Figure 8a, misalignments in damage indices can be noticed. However, these differences seem to be only really obvious for the group of paths that exhibit generally high amplitudes (i.e., A2B2, A1B1, A3B2, A3B3), in the PHASE I experimental test. Similar to the observation made for the A4B3 path (Figure 7a), undesired effects appear to obscure the course of the damage index at the beginning of the PHASE II experimental test in Figure 8a. Those effects are particular apparent for the A3B3, A2B3 and A3B2 wave propagation paths from 137,500 to 175,000 fatigue cycles. The continuing reduction of the DI for the A3B3 path in this range, despite the constant crack length, can probably be explained by a subsurface crack growth or another form of material deterioration affecting wave propagation. However, the DIs for the A2B3 and A3B2 paths demonstrate an increase in amplitude in this particular range of cycles (i.e., 137,500–175,000 cycles). This effect is difficult to explain. A reversible ageing effect of the piezoelectric material—described in transducers (*Physik Instrumente GmbH & Co. KG*, 2008, Karlsruhe, Germany)— is a possible explanation. However, this needs further investigation. The DIs of all the LEFT paths remain constant after 175,000 cycles until 380,000 cycles, which is beyond the point (approx. 335,500 cycles) when the upper left crack continued growing again. Clearly, the continuous growth of the upper left crack (after 335,500 cycles), for a duration of circa 40,000 cycles, is worryingly not reflected by any DI. A further reduction in the amplitude is exhibited only by the A3B3 characteristic at about 380,000 cycles. At about 400,000 cycles, the DIs for the A2B3 and A3B2 paths commence detecting the crack. This instance almost coincides with the crack intersecting the A2B3 and A3B2 paths, indicated by the vertical line labelled as “A2B3”. Furthermore, these characteristics demonstrate an ideal monotonically decreasing behaviour until the end of the experiment. Following the transducer layout, the A2B2 path, followed by A1B2 and A2B1 paths, are supposed to indicate a reduction in the DI at the correspondingly labelled vertical lines. The A2B2 path shows a steep drop after 445,500 cycles, which is in good agreement with crack propagation, despite unexplainable oscillations beforehand. A similar unexpected increase in the DI is exhibited by the result for the A1B2 path. However, the A1B2 and A2B1 paths demonstrate a considerable reduction in peak-to-peak amplitude after 453,500 cycles, allowing the conclusion that they detect the crack. For some unapparent reason, the DI of the A1B1 path seems to outperform those of the A1B2 and A2B1 paths, as it shows a nice monotonically declining characteristic, starting from 437,500 cycles, while the A1B2 and A2B1 paths exhibit the previously mentioned oscillations.

Similar to the observation made for the DIs of the LEFT paths, the majority of characteristics for the RIGHT paths remain relatively constant in Figure 9a. Moreover, the A4B4 path, which is the analogue of the A3B3 path but on the right of the central rivet, exhibits equally increasing peak-to-peak amplitude before it starts to decline. However, in contrast to the LEFT paths, a correlation between wave propagation length and signal strength (i.e., amplitude) cannot be found, as even diagonal paths with slightly longer propagation lengths demonstrate a high level of peak-to-peak values. In fact, the DI for the A5B4 path appears to have by far the highest level of amplitude from all the paths used in the experiment, surmounting the amplitudes of the A4B4 and A4B5 paths. In the analysis so far, almost no, or only very small, deviation between the two data sets, taken for each acoustic path and damage state, could be observed. This observation, however, is different for the DIs for the A4B4 and A4B5 paths at 100,000 cycles, as the two respective measurements demonstrate a considerable deviation from each other by almost 10% (Figure 9a). The reason for these considerable deviations could not be established. An offset between all the DI traces in the transition from the PHASE I to the PHASE II experimental tests in Figure 9a can be noticed, as was discernible for the LEFT paths. However, the offsets between the characteristics seem to be rather consistent, showing an almost proportional reduction with respect to the value of amplitude. The analysis of the peak-to-peak data for the PHASE II experimental test in Figure 9a (right column) seems to be rather complicated. Firstly, the DI for the A5B4 path significantly decreases monotonically after 285,500 cycles, even though the crack remains constant for nearly another 100,000 cycles (i.e., up to 381,500 cycles). Secondly, the peak-to-peak values for the A4B4 path begin to decline at the same cycle count, but they behave highly non-monotonically because of an incomprehensible zigzag pattern in the course of the presented characteristic. Thirdly, the graphs for the A5B6 and A6B6 paths remain constant with no indication of damage, almost until the end of the experiment, before they synchronously exhibit an even more volatile behaviour than observed from the characteristic for the A4B4 path. The supposedly premature detection of damage from the results for the A5B4 and A4B4 paths can be explained by unnoticed or subsurface crack growth beginning at 285,500 cycles. However, the highly volatile course of the DIs spoils the analysis of the A4B4 path at least after 381,500 cycles. Equally questionable and less useful are data obtained for the A5B6 and A6B6 paths after 445,500 cycles.

In summary, the peak-to-peak value of acoustic waveforms generally demonstrates a good correlation with the damage size, i.e., the DI decreases with increasing crack length. Particularly good results were obtained for short propagation lengths, when the crack intersects the direct wave propagation paths. The acoustic signals acquired from short propagation paths showed better performances because of their initially higher signal strength and hence better ratio between the acoustic signal and the noise (SNR) present in measurements. Despite the fact that the peak-to-peak amplitude seems to be a promising indicator that could be used for damage detection, two undesired effects have been observed. Firstly, the results exhibit considerable offsets between the PHASE I and PHASE II experimental measurements. This effect was rather less obvious for the central paths. Nevertheless, those offsets only demonstrate that waveform amplitudes are very sensitive to measurement conditions that were inevitably different eight years apart (because of, for example, ageing of transducers and connections, environmental effects and different data experimental equipment used), as expected. Secondly, most transducer combinations exhibited increased values of DI before expected attenuations resulting from damage. The reason for this behaviour is further investigated in Section 5.

### 4.3. Instantaneous Amplitude 

The peak-to-peak value estimation involves the extraction of the global maximum and minimum of the entire Lamb wave response. The question is why the peak-to-peak value sometimes exhibits an undesired unsteady behaviour, as for example the DI characteristics—based on the A3B4 and A4B3 wave propagation paths (Figure 7a between 105,000 and 135,000 fatigue cycles, and after 330,000 cycles). Figure 10 sheds some light on this particular issue, as the DI based on the Lamb wave response for the A3B4 wave propagation path (Figure 10a) is contrasted with the time coordinates of the extracted maximum and minimum (Figure 10b). This analysis—equivalent to the time-of-flight based on respective extreme values—demonstrates very nicely that the order in which the maxima and minima are extracted constantly changes throughout the experiment. This issue is particularly relevant to the behaviour exhibited in Figure 10b around 120,000 and after 335,000 fatigue cycles, where large disturbances can be observed. One possible solution to this problem is the analysis of the instantaneous amplitude (envelope function). This instantaneous characteristic was calculated for all Lamb wave responses following the procedure described in Section 3. The maximum of the instantaneous amplitude was calculated for all Lamb wave responses. Figure 7b, Figure 8b and Figure 9b give the results, the DI characteristics based on the maximum of the instantaneous amplitude.

As the instantaneous amplitude only generates a one-sided (positive ordinate) envelope function, the correspondingly extracted maximum values should generally only represent half the magnitude of the corresponding peak-to-peak values. Figure 7b is indeed a scaled version of Figure 7a with a scale factor of 0.5. Even in the area of transition at 137,500 cycles, no differences (apart from scaling) can be observed. This finding can be extended to the PHASE II experimental test, as no new results or insight are gained from those diagrams. Equally to the central paths, the DI based on the maximum instantaneous amplitude on the left (Figure 8b) and right (Figure 9b) paths just embodies scaled versions of the respective peak-to-peak DI for all the phases of the experiment. For the left path, this outcome can be observed in Figure 8b and compared with Figure 8a. The respective figures for the right paths are displayed in Figure 9b and can be compared with Figure 9a.

The instantaneous amplitude was integrated over the entire analysed time interval given by the length of the Lamb wave response. The resulting quantity—related to the energy of the analysed signal response—provides a new DI. The results—presented in Figure 7c, Figure 8c and Figure 9c—show that the integration inevitably contracts the range of DI values; in other words, the DI graphs become flatter if compared with the previously analysed characteristics. Another undesired change is the amplification of the non-monotonic behaviour for almost every acoustic path in all phases of the experiment. A rather mixed outcome was achieved in the transition from the PHASE I to the PHASE II experimental test, as offsets for some paths were reduced while others were increased. The DI developed in this subsection appeared to partly reduce the extent of some unexplainable phenomena in the course of the DIs, beyond the aforementioned contraction of values. Particularly obvious was this effect at the beginning of PHASE II, where the peak in A4B3, and the steep inclination and declination of the DI for the A2B3 and A3B3 paths, respectively, were curtailed. Another improvement seemed to appear in the shape of the DI characteristic, as the A4B4 path demonstrates a much better correlation to damage size at the beginning of PHASE II. Moreover, the A3B3, A2B3 and A3B2 paths appeared to indicate the continuation of the crack growth (above 340,000 cycles) much sooner than previous DIs. Rather incomprehensible phenomena were the premature indication of crack growth for the A5B6 path and the appearance of an additional peak (outlier) for the A5B4 path.

In summary, the analysis of instantaneous amplitude demonstrates no major impact on the course of the DI graphs whatsoever. The only difference is in the scaled magnitude of the new DI, as the extracted maximum only represents half of the peak-to-peak value. The instantaneous amplitude integration has led to mixed results. On the one hand, this new amplitude measure demonstrated improvements in terms of a better correlation to damage, e.g., the result for the A4B4 path. On the other hand, this integration amplified the non-monotonic characteristics in DIs.

### 4.4. Novelty Detection

Outlier analysis was undertaken to investigate the deviation of extracted features from their normal condition and illustrate how statistical analysis—based on diffuse Lamb wave data—could be used in practice for crack detection. The work presented is based on the novelty detection approach used in [56]. The word outlier refers here to data irregularities that could be linked to structural damage. Assuming univariate data, this irregularity measure can be determined as,
(5)$$z_{\xi }=\frac{\left|x_{\xi }-\mu_{x}\right|}{\sigma_{x}}$$
where, $x_{\xi }$ is the test sample (potential outlier) and $\mu_{x}$, $\sigma_{x}$ are the sample mean and standard deviation of the data set, respectively. This measure is compared against an objective criterion to determine whether a data sample is unlikely or likely to be an outlier. The determination of the threshold value that classifies the tested observation as an outlier is crucial in this analysis. A Monte Carlo approach is used here to construct the 99% threshold, that for univariate data corresponds to approximately 2.576 standard deviations away from the mean, which can be considered as a significant deviation.

The outlier analysis was performed for the A3B4 Lamb wave responses as an example. The results for the peak-to-peak value, maximum envelope and integrated envelope are given in Figure 11a–c, respectively. The outlier analysis of these parameters reveals no false positives, confirming their validity in this statistical approach. The results also show that the clearance between the threshold and the outliers exhibits the largest value for the integrated envelope parameter, indicating the best statistical confidence with which an actual outlier (fatigue crack) is identified. The analysed parameters increase with the crack length and therefore could potentially be used for the estimation of damage severity, given an appropriate means of calibration.

## 5. Simulation of Wave Diffraction from a Crack Tip

The amplitude analysis of the Lamb wave responses presented in Section 4 has demonstrated that short direct wave propagation paths exhibited an excessive level of damage index oscillations at the beginning of PHASE II of testing, before expected attenuations, resulting from the damage. Since the undesired effect exists when the actuator-sensor path crosses the tip of the propagating crack, this behaviour has been further investigated using numerical simulations. It is important to underline that the simplified modelling approach was performed not to validate the experimental wave propagation results but only to explain the undesired—and above mentioned—damage index effect.

A simple wave propagation model was used. The focus of this modelling effort was on the simulation of Lamb wave response changes resulting from advancement of a bisecting crack, perpendicular to the acoustic propagation path since the other—i.e., CENTRAL wave propagation paths—do not cross the crack The relevant model is illustrated in Figure 12. The Local Interaction Simulation Approach (LISA)—presented in [57,58]—was implemented for this simulation. MATLAB software based on the semi-analytical Lamb wave propagation solution [59] was used to validate numerical simulation results.

Despite the use of a flat 5-cycle sine burst excitation signal in the experimental configuration, a Hann-windowed 5-cycle sine burst was chosen for the simulation because of its more narrow-band frequency characteristics. The assumed medium of the wave propagation was a 2 mm thick aluminium plate. The velocities of the longitudinal and shear waves were set up as v_L_ = 6420 m/s and v_T_ = 3040 m/s, respectively. The subsequent simulation was solely based on out-of-plane responses. A three-dimensional LISA model was required to simulate changes in Lamb wave response resulting from the advance of the crack perpendicular to the wave propagation direction in the plane of the plate, following the geometrical boundary conditions encountered in the experiment for the short (direct) paths such as A1B1, A2B2, A3B3, A4B4 and so forth. The other configurations, such as the short diagonal (e.g., A3B4) and the long diagonal paths (A6B1, A1B6), were neglected in the modelling effort, as the chosen model was considered to be sufficient to capture major effects affecting the courses of the envisaged DIs. The semi-infinite crack was modelled as a notch with a width of one element (cell). Figure 12 demonstrates the simple geometry of the selected model and defines the parameter crack-tip position, “a,” which has its point of origin at the line-of-sight between the actuator and receiver. This definition proved to be practical as the position a = 0 indicates the beginning of the intersection of the acoustic path by the crack. The parameter range from −40 to +40 mm was chosen for two reasons. Firstly, a crack-tip position 40 mm away from the line of sight (i.e., a = 40 mm) seemed to be just far enough to not affect the wave propagation, and secondly, the 40 mm distance corresponds to the horizontal spacing between the considered transducer pairs, granting a perfect overlapping of the areas of inspection. The Lamb waves were simulated for a time interval from 0 to 50 μs, as this was sufficient to capture the arrival of the first wave packet throughout the entire range of different crack-tip positions. Once the model was discretised, a convergence analysis was performed to achieve the desired accuracy of the model. As a result, four cells were used through the thickness of the plate in all numerical simulations. This level of discretisation was sufficient to obtain good amplitude convergence of the LISA-based results with the semi-analytical solutions.

Figure 13 summarises all numerical simulation results. Here, simulated (right vertical axis) and experimental (left vertical axis) DIs—based on peak-to-peak values—are compared for various crack-tip positions. The constant trend of the DI simulated characteristics in Figure 13 between −40 and −25 mm demonstrates that the analysed crack-tip positions do not affect Lamb wave propagation at all. For that reason, the entire graph was normalised by the result of the model for a = −40 mm, permitting the analysis relative to a presumably pristine condition (i.e., no crack condition). Figure 13 unambiguously demonstrates an increasingly oscillatory behaviour of the DI (between a = −25 and a = −5) before the crack intersects the acoustical path (the line-of-sight) and the peak-to-peak values expectably reduces because of the presence of the crack. The graph at those oscillations reveals amplitudes exceeding the initial values, i.e., DI > 1. This oscillatory behaviour is the result of constructive and destructive interference between the wave diffracted from the crack and the wave propagating alongside the direct path (Figure 12). The simulation results are in relatively good agreement with the experimental behaviour observed in the standard aircraft splice joint. The graphs for the A3B3 and A4B4 wave propagation paths in Figure 13 demonstrate unambiguously an increased peak-to-peak value when the crack approached the acoustic path. This behaviour is less obvious for the A2B2 wave propagation path. However, the DI graph for the same propagation path—but plotted versus number of cycles—in Figure 8a indicates an increased peak-to-peak value before the crack intersects the acoustical path. 

In summary, simple numerical simulations proved to be an invaluable addition to the presented work, and were sufficient to explain undesired oscillations of DIs in the experimental results. These oscillations resulted from positive and negative interactions between the incident and diffracted wave components.

## 6. Discussion and Further Comments

This section discusses further major findings with respect to the experimental procedure, signal processing and crack detection results. Most comments are given with respect to the monitoring strategy, implementation and possible engineering application. 

Although SMART Layers^®^ were used for crack detection in the research work presented, other transducers—discussed in [8]—could be used for Lamb wave actuation and sensing. The major advantage of the SMART Layers^®^ is that such transducers are low-profile and can be easily bonded on the structure. When composite panels were tested (e.g., impact damage detection) these transducers could even be embedded. Thus the benefit for aerospace application is the major advantage of the transducers used.

The operation of the SMART Layers^®^ piezoelectric transducers was quite remarkable. Despite the fact that the panel was left for eight years after the first phase of experimental tests, the majority of transducers performed quite well. Some detrimental ageing effects (e.g., possible depolarisation of the piezoceramic element and/or deterioration of the coupling layer) were observed only for one transducer.

Although a relatively low level of voltage amplitude (10 V) was used for Lamb wave excitation, high SNR Lamb wave responses have been acquired from the transducers. Shorter wave propagation distances have led to less attenuation, as expected. However, for the complex structure investigated (splice joint panel with complex geometrical features such as adhesive layers and multiple rivets), even relatively low excitation amplitudes were sufficient to obtain good-quality Lamb wave data across distances longer than 200 mm. This confirms that high-voltage excitations are not required when cracks are monitored in complex structures. Since most previous applications in the field required high-voltage excitation [8], this finding is also good news to the aerospace community.

It is important to note that the proposed maximum-likelihood estimate—used to obtain information on SNR levels—was effective to assess the quality of experimental Lamb wave data.

The selection of the excitation frequency is essential when Lamb waves are used for structural damage detection. The excitation frequency decides on the wavefield complexity and damage detection sensitivity. In practice, the frequency selection in Lamb-wave-based damage detection is always a compromise. On the one hand, the lower the frequency, the less complex (smaller number of propagating modes) the Lamb wave field needs to be in order to be analysed. On the other hand, the higher the frequency, the better the sensitivity of the damage detection. However, the proposed methodology is based on the diffuse Lamb wave field. This approach does not require physical understanding of wave propagation and modelling of dispersion characteristics that lead to careful selection of the excitation frequency. The excitation frequency can be selected arbitrarily. The major advantage of this approach is its simplicity, extending the range of possible applications to a much wider spectrum of complex materials and structures. This is due to the fact that the intricate identification of Lamb wave modes via complicated and often unavailable dispersion characteristics is not required. The only limitation of the excitation frequency relates to the damage detection sensitivity. A rule of thumb in ultrasonic applications is that discontinuities that are larger than one-half the size of wavelength can usually be detected. However, previous studies show that the sensitivity of Lamb waves is not a simple function of wavelength. The ability of guided waves to detect defects much smaller than the wavelength is widely reported in the literature. This problem is well-discussed in [8].

The amplitude-based DIs investigated—i.e., peak-to-peak value, maximum of the envelope and the integrated envelope—have produced very similar damage detection results. These results show that amplitude-based DIs decrease because of wave attenuation caused by damage. However, these indices can also increase initially due to positive interactions between incoming waves and waves diffracted from damage. This experimental effect—confirmed by simple numerical simulations—can also be used for crack detection.

The results show that, when amplitudes of direct Lamb wave responses are used for damage detection, transducers do not need to be positioned in the line-of-sight regarding the damage. In other words, wave propagation paths do not need to cross cracks physically. Some indication about damage can be obtained even 20 mm before the crack intersects the wave propagation paths. This observation extends the potential area of inspection and reduces the number of transducer pairs required for damage detection. However, the best damage detection results were obtained when wave propagation paths were crossing through the centre of damage. In such situations, shorter wave propagation paths gave earlier indications of damage.

The novelty detection approach was undertaken to demonstrate that statistical analysis and crack detection confidence levels are important for any future implementation and practical engineering applications. The novelty detection approach used demonstrates that the univariate analysis based on amplitude-related parameters can be used effectively, not only for damage detection, but also for the assessment of damage severity. The integrated envelope function is statistically the most confident parameter investigated that could be used to fulfil these tasks. However, much deeper analysis is needed to confirm this finding. The threshold definition in novelty is crucial for good reliability of the method. The effects of changing the threshold are obvious from a consideration of Figure 11. Increasing the threshold from 99% confidence would eliminate the false positive, but introduce no false negatives. There would always be a trade-off between false positives and false negatives in raising the threshold. In that case, a higher threshold is arguable as it would eliminate false alarms at the small cost of some sensitivity.

The proposed method is baseline free. The experimental work presented here indicates that Lamb wave amplitudes can be used effectively for crack detection only when template data representing the undamaged condition are available. If crack detection is an issue, data representing the undamaged condition are needed. If crack propagation is a problem, some reference data referring to the current crack length are needed. 

Finally, it is important to underline that two fatigue and damage detection tests performed eight years apart, in completely different experimental conditions, produced good experimental results. The application of different fatigue machines, different specimen clamping, different experimental equipment, different environmental conditions and different excitations (with and without the envelope) produce remarkably good crack detection results with the diffuse Lamb wave field used and the relatively simple data processing approach undertaken. The overall performance of the transducers and crack detection capability was maintained. As a result, the crack could be monitored effectively.

## 7. Conclusions

Diffuse Lamb wave responses have been used for fatigue-crack detection in aluminium riveted lap-joint panels. Surface-bonded SMART Layers^®^ transducers have been used in this study for Lamb wave excitation and sensing. The experimental work involved a number of wave propagation paths and different crack lengths.

The results show that the amplitude of diffuse Lamb wave responses—via peak-to-peak, maximum envelope and integrated envelope values—can be used effectively for crack detection in complex multi-riveted aerospace structures. A clear link and good correlation between damage indices used and crack lengths observed has been found. 

The major advantage of the proposed method is its simplicity. In contrast to most previous applications, the intricate identification of Lamb waves propagating modes via complicated—and often not possible to obtain—dispersion curves is not required. Thus the range of application could be extended to a much wider spectrum of complex materials and structures when the method is used.

The implementation and future engineering application of the proposed method would require further research work that should focus in particular on statistical analysis and crack detection confidence levels. Any future work could also focus on phase/frequency analysis of Lamb wave responses and methods that can compensate for possible data trends related to operational/environmental conditions. Potential application of the method to composite structures (e.g., impact damage detection) could also be investigated.

## Figures and Tables

**Figure 1 materials-16-01619-f001:**
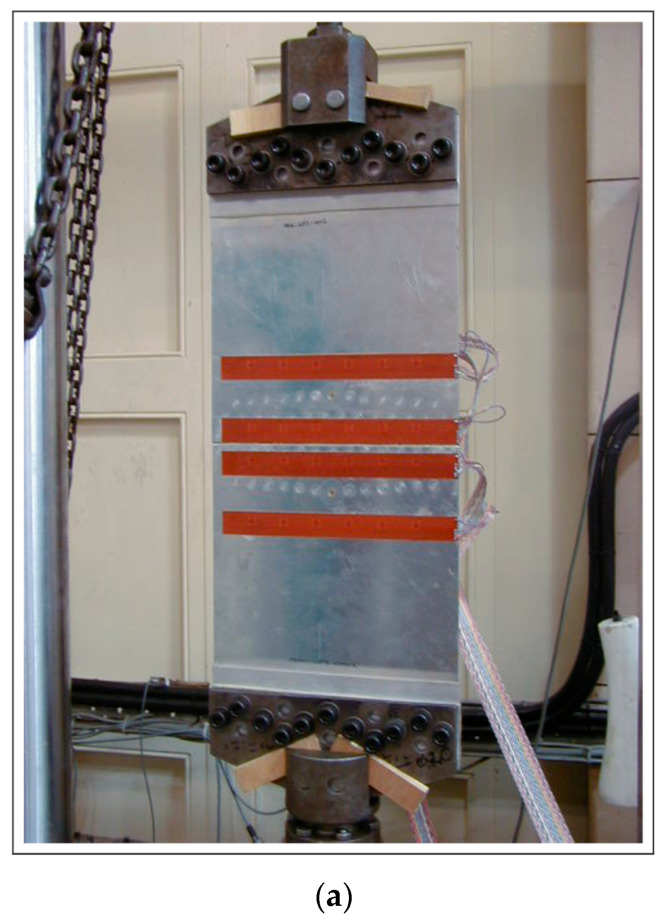
Multi-riveted strap joint aluminium panel with bonded A, B, C and D SMART Layer^®^ transducers: (**a**) general view; (**b**) geometry and dimensions; circle K gives transducer locations that are given in details in Figure 2.

**Figure 2 materials-16-01619-f002:**
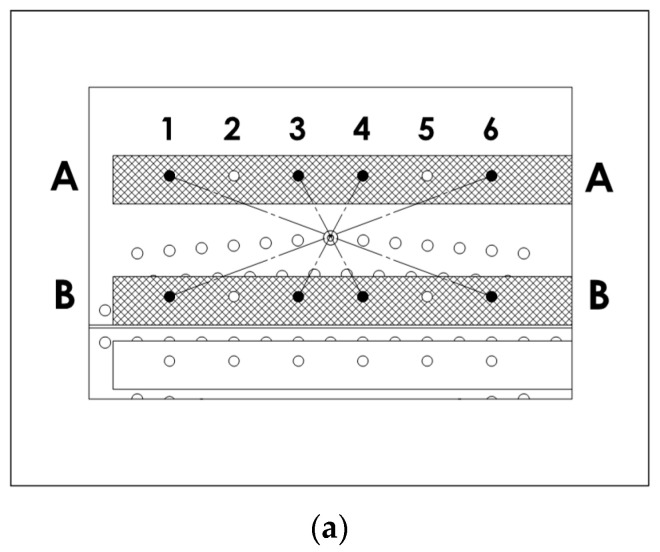
SMART Layer^®^ transducer locations and wave propagation paths: (**a**) paths crossing the central rivet—referred to as CENTRAL paths; (**b**) paths on the LEFT of the central rivets; (**c**) paths on the RIGHT of the central rivet. A and B indicate two SMART Layers^®^ used.

**Figure 3 materials-16-01619-f003:**
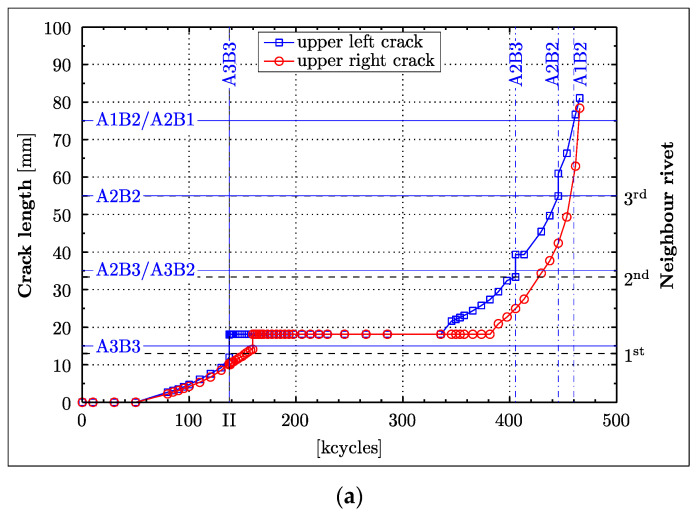
Crack propagation curves. The thin vertical solid line (described by the symbol “II”) separates the two experimental tests (PHASE 1 and PHASE 2) investigated. The other lines indicate at which point in time (or after how many fatigue cycles) the cracks intersected the respective wave propagation paths. These other lines are plotted for: (**a**) LEFT wave propagation paths; (**b**) RIGHT wave propagation paths.

**Figure 4 materials-16-01619-f004:**
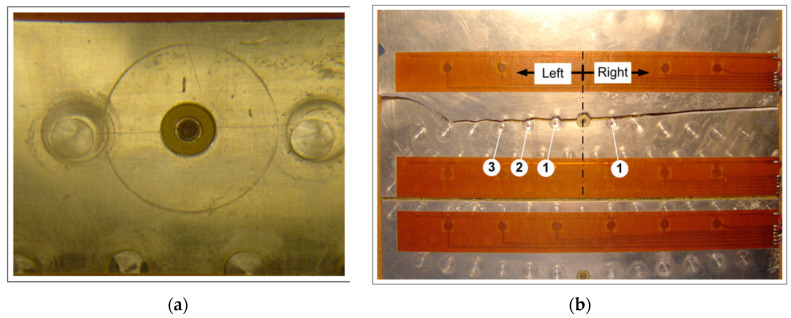
Monitored fatigue crack: (**a**) after 347,500 fatigue cycles; (**b**) completely fractured panel after 465,500 fatigue cycles. Numbers indicate consecutive rivets to the left and right side from the central rivet.

**Figure 5 materials-16-01619-f005:**
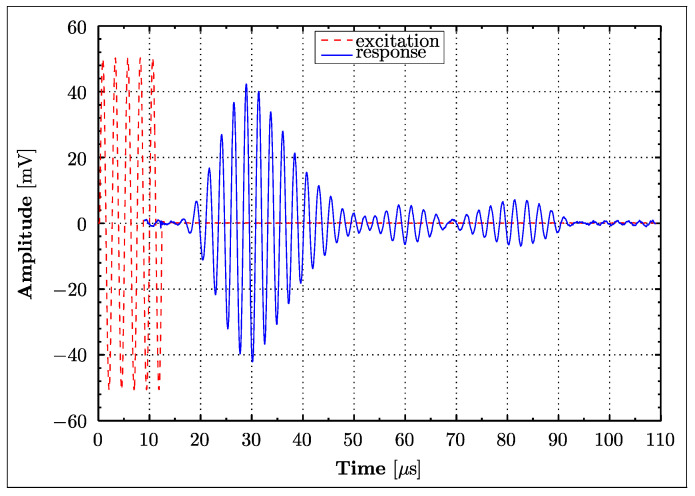
Excitation signal (scaled 1:100) and an example of the Lamb wave response for the A5B4 wave propagation path (undamaged condition).

**Figure 6 materials-16-01619-f006:**
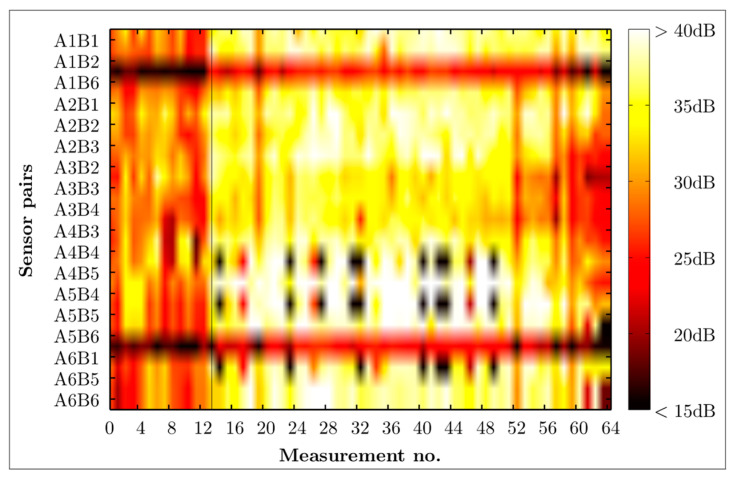
Matrix of SNR estimates for all transducer combinations in the AB layout AB. The vertical thin black line indicates the separation of Phase 1 and Phase 2 experimental tests.

**Figure 7 materials-16-01619-f007:**
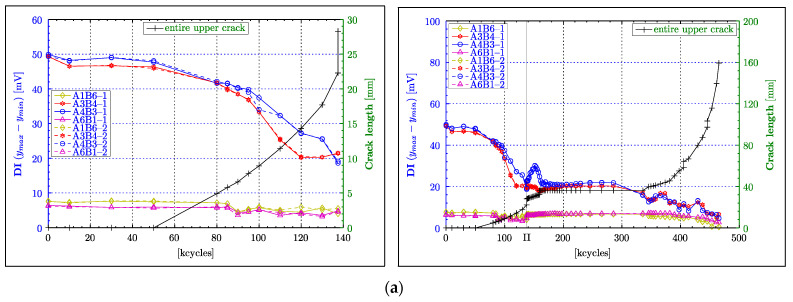
Crack detection results for the CENTRAL wave propagation paths. Damage indices are based on: (**a**) maximum peak-to-peak amplitude; (**b**) maximum instantaneous amplitude (envelope); (**c**) integrated instantaneous amplitude. The left column gives the results corresponding to only the PHASE 1 experimental test (from 0 to 147,500 fatigue cycles only) whereas the right column presents the results for the PHASE 1 and PHASE 2 experimental tests. The thin vertical solid line (described by the symbol “II”) separates the two experimental tests (PHASE 1 and PHASE 2) investigated.

**Figure 8 materials-16-01619-f008:**
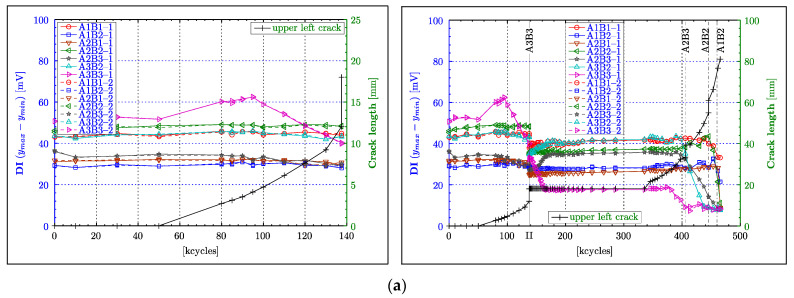
Crack detection results for the LEFT wave propagation paths. Damage indices are based on: (**a**) maximum peak-to-peak amplitude; (**b**) maximum instantaneous amplitude (envelope); (**c**) integrated instantaneous amplitude. The left column gives the results corresponding to only the PHASE 1 experimental test (from 0 to 147,500 fatigue cycles only) whereas the right column presents the results for the PHASE 1 and PHASE 2 experimental tests.

**Figure 9 materials-16-01619-f009:**
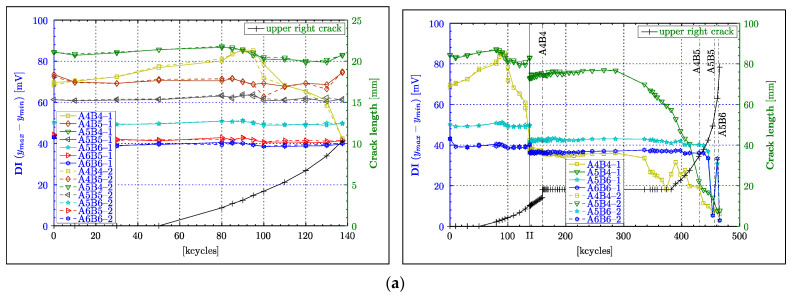
Crack detection results for the RIGHT wave propagation paths. Damage indices are based on: (**a**) maximum peak-to-peak amplitude; (**b**) maximum instantaneous amplitude (envelope); (**c**) integrated instantaneous amplitude. The left column gives the results corresponding to only the PHASE 1 experimental test (from 0 to 147,500 fatigue cycles only) whereas the right column presents the results for the PHASE 1 and PHASE 2 experimental tests.

**Figure 10 materials-16-01619-f010:**
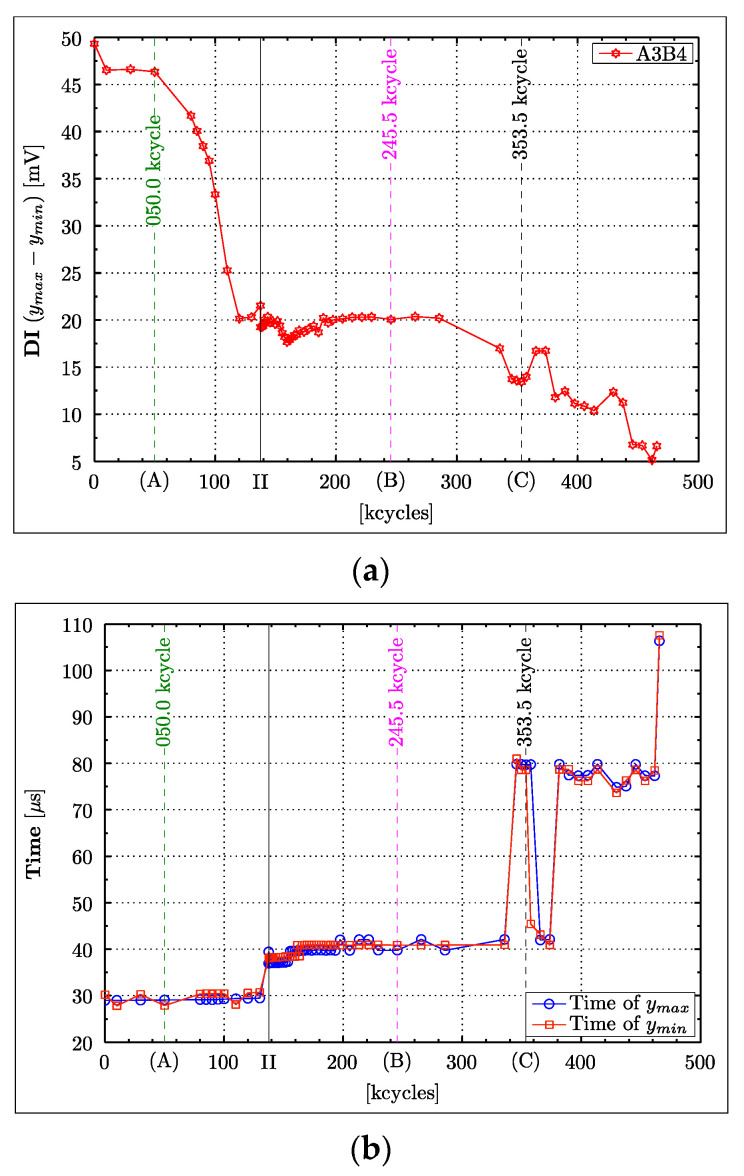
Extraction of peak-to-peak values for the A3B4 wave propagation path for consecutive fatigue cycles: (**a**) damage index based on the peak-to-peak value; (**b**) temporal location of respective maxima and minima used for peak-to-peak amplitude evaluation.

**Figure 11 materials-16-01619-f011:**
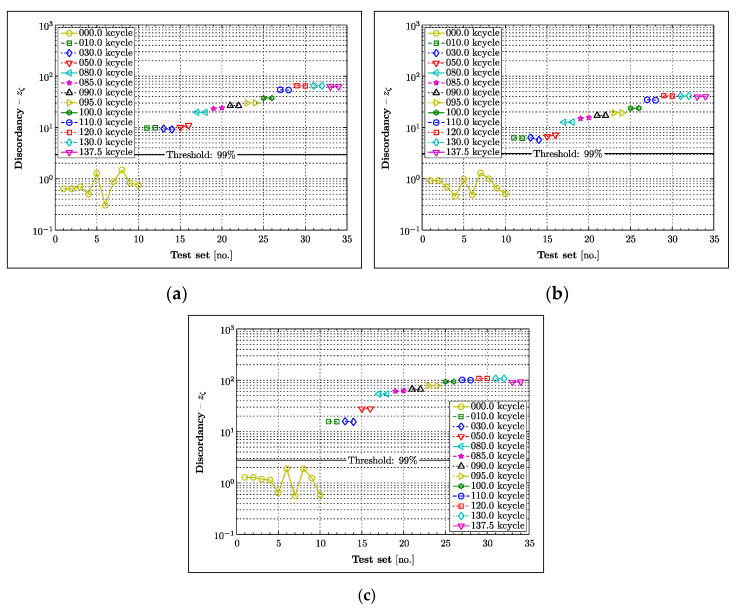
Univariate outlier analysis for analysed damaged indices based on: (**a**) peak-to-peak value; (**b**) maximum of envelope function; (**c**) integrated envelope function.

**Figure 12 materials-16-01619-f012:**
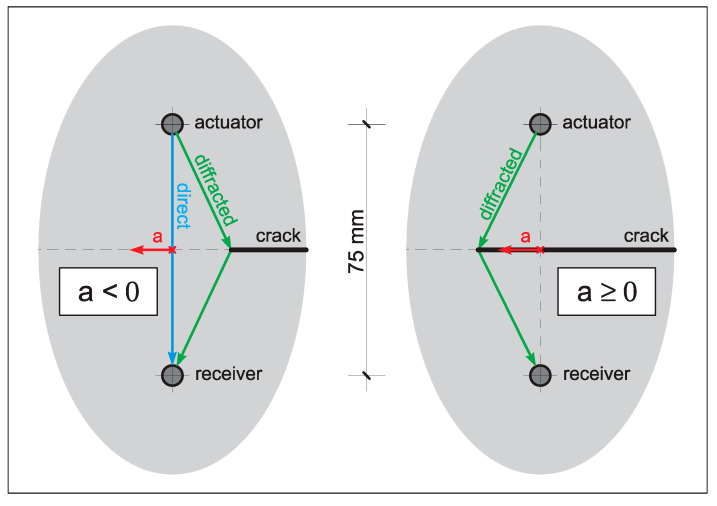
Wave propagation model near the crack tip—short (direct) and long (diffracted) wave propagation paths—for two different crack lengths, illustrating how the amplitude of the response wave is affected by diffraction.

**Figure 13 materials-16-01619-f013:**
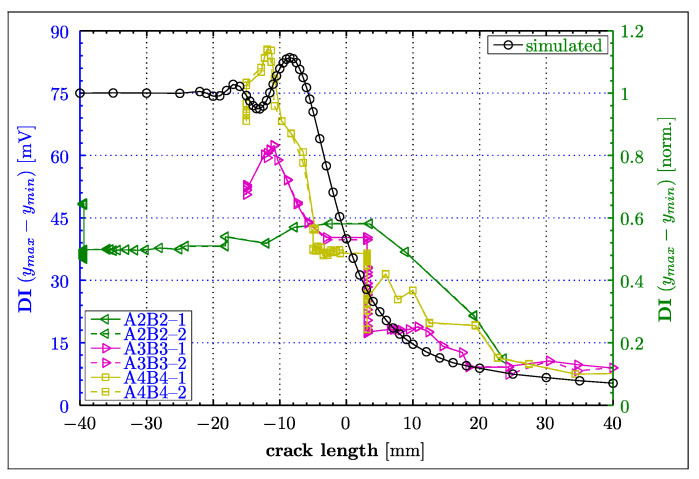
Experimental and simulated damage indices based on the peak-to-peak values for various crack tip positions “a” defined in Figure 11. Left and right vertical axes are given for the experimental and simulated results, respectively.

## Data Availability

Data available on request.

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
