# Peer review of "Fatigue-Crack Detection in a Multi-Riveted Strap-Joint Aluminium Aircraft Panel Using Amplitude Characteristics of Diffuse Lamb Wave Field"

_materials, 2023, doi:10.3390/ma16041619_

Round 1

Reviewer 1 Report

The main goal of this paper was to demonstrate potential and limitation of using amplitude characteristics of diffuse Lamb-wave Field  for monitoring fatigue cracks in a Multi-riveted strap-joint Aluminium Aircraft Panel. In that way, experimental ultrasonic testings as well as some numerical simulation studies have been conducted using pitch-catch methods. More precisely, the SMART layers transducers operate with Lamb wave modes at 410 kHz and three different damage indices based on attenuation were applied to quantify the extent of the fatigue crack. The main conclusion of this study is that a univariate analysis based on amplitude-related parameters can be used effectively, not only for damage detection, but also for the assessment of damage severity.

Overall, the paper is well written and the analysis appears carefully conducted. Although the proposed methods to detect the fatigue crack appear incremental from my point of view, the real innovation lies in the numerical simulation comparison to explain the experimental results. A more systematic qualitative and quantitative comparison between numerical simulations and experimental tests will be useful. Furthermore, no physics is given to explain the Lamb waves modes generated as well as the scattering effect. Therefore, I suggest major revisions, according to the following comments.

1) Contrarily to the experimental study, the authors have conducted simulations only for the short (direct) paths like A1B1, A2B2, A3B3, A4B4 and not with the other configurations such as the short diagonal (e.g., A3B4) and the long diagonal paths (A6B1, A1B6). From my point of view, a systematic comparison between numerical simulations and experimental tests should be made to increase the relevance of this paper.

2) More precisions should be given on the used SMART Layers transducers for the experiments, i.e. on its material characteristics and its vibration operation mode.

3) No information is given on the choice of the excitation frequency nor on the wavenumbers generated whereas these characteristics are predominant for the Lamb wave modes sensitivity to a damage.

Reviewer 3 Report

The manuscript addresses a relevant topic, Fatigue-Crack Detection in a Multi-Riveted Strap-Joint Aluminium Aircraft Panel Using Amplitude Characteristics of Diffuse Lamb-Wave Field is inside the scope of the journal, however, the following comments should be solved:

1)      The English language of the paper must be refined.

2)      In the introduction section should be improved the explanation of the problem, objectives and organization of the manuscript.

3)      Please clarify the point of novelty of the text? The subject is attractive but the method is simple.

4)      The literature review of the work must be updated.

5)      The mathematical equations are written in an unorganized, incomprehensible manner, and in an inappropriate style for the MDPI template, I recommend re -coordinating and summarizing them more than that. I also recommend inserting a Table of symbols and Notations before the introduction.

6)      The results discussion should be improved. They should be further analyzed. 

7)      Conclusion is poor and needs prominent points.

8)      References must be updated with related and modern research such as (2020, 2021, and 2022).

Round 2

Reviewer 1 Report

The authors have now mostly met the proposed comments and recommendations in the revised manuscript version, which definitely improves its quality.

Therefore, I feel this paper acceptable for publication.

Reviewer 2 Report

The author has modified according to the opinions, and I suggest accept it as its current form.